# The Effect of Adverse Mental Health and Resilience on Perceived Stress by Sexual Violence History

**DOI:** 10.3390/ijerph19084796

**Published:** 2022-04-15

**Authors:** Katherine M. Anderson, Kiyomi Tsuyuki, Alexandra Fernandez DeSoto, Jamila K. Stockman

**Affiliations:** 1Department of Behavioral, Social, and Health Education Sciences, Rollins School of Public Health, Emory University, Atlanta, GA 30322, USA; 2Department of Medicine, Division of Infectious Diseases and Global Public Health, University of California San Diego School of Medicine, San Diego, CA 92093, USA; ktsuyuki@health.ucsd.edu (K.T.); alf013@health.ucsd.edu (A.F.D.); jstockman@ucsd.edu (J.K.S.)

**Keywords:** sexual violence, depression, PTSD, resilience, stress

## Abstract

Sexual violence, including nonconsensual sexual initiation and rape, remains pervasive, with impacts including adverse mental health and dysregulated stress response. Resilience is a promising interventional target. To advance the science, we examined the potential for resilience as an interventional tool by estimating associations between resilience, adverse mental health, and perceived stress among women by sexual violence history and partner perpetration. We analyzed 2018–2020 baseline survey data from 65 women enrolled in a prospective case-control study of sexual violence and HIV susceptibility in San Diego, CA. Multiple linear regressions were performed to examine associations, stratified by sexual violence history. About half of women experienced nonconsensual sexual initiation and/or rape; half of rapes were partner-perpetrated. Post-traumatic stress disorder (PTSD) was significantly associated with perceived stress among survivors (in regressions with depression and resilience, nonconsensual initiation: β = 6.514, *p* = 0.003, R^2^ = 0.616; rape: β = 5.075, *p* = 0.030, R^2^ = 0.611). Resilience was associated with lower perceived stress for all women; the effect appeared stronger among survivors of sexual violence (nonconsensual initiation: β = −0.599, *p* < 0.001 vs. β = −0.452, *p* = 0.019; rape: β = −0.624, *p* < 0.001 vs. β = −0.421, *p* = 0.027). Partner perpetration of rape was not associated with perceived stress. Our findings support leveraging resilience and addressing PTSD to reduce perceived stress among women with lifetime experiences of sexual violence.

## 1. Introduction

Sexual violence against women remains pervasive, and is associated with long-term negative health consequences [1]. In the United States (U.S.), approximately one in five women (21.3%) experience rape in their lifetime [2], and 8.8% experience rape by an intimate partner [3], with racial and ethnic minority women disproportionately affected [4,5,6,7,8,9,10,11]. Young women face the greatest burden of sexual violence, with 79% of women’s first rape experiences occurring before the age of 25 and 40% before the age of 19 [3]. Further, nationally representative samples suggest that approximately 7% of U.S. women experience forced sexual initiation, with racial and ethnic minority women disproportionately impacted [12,13]. Of these, a large part is attributable to intimate partner violence (IPV) or domestic violence [12], wherein a former or current partner or spouse is the perpetrator [14]. While sexual violence estimates are already alarmingly high, national estimates often underrepresent the true burden, and two out of three survivors of sexual violence are expected to experience sexual violence revictimization at some point in their lifetime [15].

### 1.1. PTSD, Depression, and Stress among Survivors of Sexual Violence

Survivors of sexual violence may suffer long-lasting psychological morbidity, including post-traumatic stress disorder (PTSD) and depression [16,17,18,19,20,21]. Of traumatic experiences, rape is the most likely to be associated with PTSD, with almost one-third of survivors developing PTSD during their lifetime [22]. Additionally, one-third of sexual violence survivors experience major depression during their lifetime [23]. The relationships between depression, adverse mental health, and stress are well documented [18,24,25,26]: comorbidity of PTSD and perceived stress is higher among survivors of sexual violence than those who have not experienced sexual violence [27], and women who experience sexual IPV report more depressive symptoms, which in turn is associated with higher perceived stress, than survivors of non-sexual IPV. However, the impacts of trauma-related stress vary depending on the type of trauma experienced [28]. Survivors of sexual violence report significantly worse psychological and physical health outcomes compared to individuals without experiences of sexual violence, without experiences of trauma, and with experiences of non-sexual violence trauma [28]. Survivors of sexual violence have a heightened level of perceived stress [29], which is often measured by the lack of perception predictability or control of one’s life, high extent of life changes, and lacking ability to cope with problems or difficulties [30]. Stress, in turn, has significant negative physiological implications. Chronic and acute stress impact immune responses and wound healing [31,32,33,34,35,36,37], increase allostatic load [38], and promote chronic inflammation [39,40]. Such impacts have been documented years following trauma [41], with implications for increased susceptibility to negative health outcomes [42,43].

### 1.2. The Role of Partner Perpetration

While all survivors of sexual violence may experience adverse mental health impacts, relationship to the perpetrator of violence may also impact outcomes [44]. Intimate partner sexual violence (IPSV), sexual violence perpetrated by a current or former spouse or romantic partner, including a dating partner [44], accounts for 45% of all sexual violence in the U.S. [3], with a significant proportion occurring in the context of domestic violence. Though sometimes used interchangeably with intimate partner violence [14], domestic violence may also refer to violence in other domestic relationships, such as partner–child or sibling relationships; however, sexual domestic violence significantly overlaps with IPSV. IPSV is often overlooked as a differentially impactful experience compared to non-partner sexual violence [44], despite evidence that IPSV is more strongly associated with depressive symptoms than non-partner sexual violence [45,46]. Women who experience IPSV also report significantly higher PTSD and anxiety scores compared to women with no sexual violence experiences [46].

### 1.3. Resilience among Survivors of Violence

Promisingly, resilience—the process of successfully adapting to adversity, trauma, or significant stressors [47]—may mitigate the relationship between psychological trauma, adverse mental health, and stress [18,48], but few studies have examined these associations [49,50]. Fewer still have examined this relationship among survivors of sexual violence, despite the high co-occurring prevalence of adverse mental health and trauma-related stress [18,51,52]. One study among Black women in Baltimore, Maryland, USA found that among survivors of sexual violence, resilience partially attenuated (mediated) the association between perceived stress and severe depression [18]. While women who experience domestic violence are reported to have lower resilience scores than the general population, women experiencing IPV in the domestic context had higher resilience than those experiencing paternal violence [53], indicating a solid foundation for resilience-based intervention work. Investigation of resilience, adverse mental health, and stress among sexual violence survivors, with consideration for type of sexual violence experienced and perpetrator of violence, is, therefore, critically needed.

To address this gap in research, we sought to model the associations between symptoms consistent with PTSD, depression, and resilience with perceived stress among survivors of nonconsensual sexual initiation and lifetime rape, with consideration for partner perpetration. Findings from this research may provide support for the development of trauma-responsive interventions to support survivors of sexual violence.

## 2. Materials and Methods

### 2.1. The THRIVE Study

The current analysis used baseline data from a sample of 65 participants aged 18 and older who enrolled from January 2018 to December 2020 into The THRIVE Study. Detailed methods for The THRIVE Study have been published elsewhere [42]. In brief, The THRIVE Study was a prospective case-control study designed to examine the impact of sexual violence on mental health and immune and stress response dysregulation, with implications for HIV susceptibility, among adolescent girls and adult women residing in San Diego, CA, USA. Participants were recruited through community organizations, a local rape crisis center, physical flyers, and social media advertisements. Eligibility criteria included being between the ages of 14 and 45, currently residing in San Diego County, CA, self-report being cis-gender female, and self-report having either experienced in the past month: (1) consensual vaginal sex with a male partner (controls) or (2) nonconsensual vaginal penetration perpetrated by a male (cases). Only women aged 18 and older expressed interest in the study. Eligible and interested participants provided written informed consent at a baseline study visit and attended two additional study visits over the course of three months, resulting in three study timepoints (Baseline, 1-Month Follow-Up, 3-Month Follow-Up). At each study visit, participants completed an interviewer-administered survey (30–60 min), pregnancy testing, blood draw for HIV testing, and a cervicovaginal exam performed by a female physician or nurse practitioner to collect (1) swabs for STI and bacterial infection testing, and (2) cervicovaginal lavage fluid for assessment of immune biomarkers local to the female reproductive tract. Following each visit, participants also self-collected nine saliva samples over the course of three consecutive days (three timed samples collected per day), which were then retrieved by study staff. Participants received $50 in compensation for each study visit and an additional $35 upon return of saliva samples; transportation assistance was provided in the form of complimentary rideshare services. All participants were also provided with a list of local resources for free or low-cost medical and social services as well as facilitated connection to services when requested. Robust protocols were employed to protect the safety and comfort of all participants, inclusive of integration of Trauma-Informed Care [54,55], screening for suicidality using the Suicide Behaviors Questionnaire (SBQ-R) [56], and screening for risk of homicide by an intimate partner using the Danger Assessment [57]. All procedures were approved by the University of California San Diego Institutional Review Board prior to enrollment of participants.

### 2.2. Measures

#### 2.2.1. Mental Health Indices

The Perceived Stress Scale (PSS) [30] was used to measure perceived stress. The PSS consists of 10 items, each measuring the degree to which situations in one’s life are appraised as unpredictable or uncontrollable. Example items included being upset because of something that happened unexpectedly and feeling unable to control the important things in one’s life. Responses were coded on a 5-point Likert scale ranging from 0 (never) to 4 (very often). Scores were summed, with higher scores indicating higher levels of perceived stress (sample Cronbach’s alpha = 0.974).

PTSD was measured using the Primary Care PTSD Screen (PC-PTSD-4) [58]. Participants were asked four questions about how a traumatic event over the course of their life affected them over the past week. Example items included having had nightmares about the traumatic event and being constantly on guard, watchful, or easily startled; response options were “yes” or “no”. Scores were summed, and participants endorsing “yes” for three or more items were classified as having symptoms consistent with PTSD. Cronbach’s alpha for the PC-PTSD-4 scale was good and acceptable (sample Cronbach’s alpha = 0.721) [59]. Resilience was measured using the 10-item Connor–Davidson Resilience Scale (CD-RISC) [60]. Participants were asked questions that measured how well one is equipped to bounce back after stressful events, tragedy, or trauma in the past week. Responses were coded on a 5-point Likert scale ranging from 1 (not true at all) to 5 (true nearly all of the time). Scores were summed, with higher scores indicating higher levels of resilience (sample Cronbach’s alpha = 0.888). Depression was measured using the 10-item Center for Epidemiological Studies Depression Scale (CES-D-10) [61]. The CES-D-10 is a self-report measure of depressive symptoms assessed in the past week using a 4-point Likert scale ranging from 0 (rarely or none of the time) to 3 (all of the time). Example items included being bothered by things that do not normally bother you, feeling depressed, and feeling that everything was an effort. Two positive affect statements were reverse coded. Scores were summed, with higher scores indicating higher levels of depressive symptoms (sample Cronbach’s alpha = 0.923).

#### 2.2.2. Sexual Violence History

Sexual initiation type was measured by asking “How would you describe your first [vaginal/anal/oral] sexual experience?” with the following response options: (1) wanted and not forced, (2) wanted but pressured, (3) unwanted and pressured, (4) unwanted and threatened with violence, (5) unwanted and physically forced, or (6) unwanted and forced to drink alcohol or take drugs. Nonconsensual sexual initiation included participants who endorsed any category between 2 and 6. Ever experiencing rape in their lifetime was assessed by asking participants the number of times a male or female partner or non-partner had used force or threats to make the participant have sex. Participants who endorsed > 1 times were classified as ever experiencing rape; those who endorsed 0 times were classified as never experiencing rape in their lifetime. Among those who ever experienced rape, an additional variable was created to capture whether or not they had experienced rape perpetrated by an intimate partner. 

Of note, participants who endorsed a nonconsensual sexual initiation were not automatically classified as having experienced rape; rather, the measure for rape was based on self-defined experience of forced or threatened sexual activity. As such, neither variable is mutually inclusive of the other.

### 2.3. Statistical Analysis

Statistical analyses were performed in SPSS version 26 [62]. We assessed differences in symptoms consistent with PTSD, depression, and resilience on perceived stress stratified by sexual violence history. There were no missing data. We computed descriptive statistics for all variables, reporting means and standard deviations for normally distributed continuous variables, medians and interquartile range for non-normally distributed continuous variables, and frequencies and proportions for categorical variables. We conducted bivariate analyses using independent sample *t*-tests, chi-squared tests, Fischer’s exact tests, and Pearson correlations. Multiple linear regressions were used to examine the influence of symptoms consistent with PTSD, depression, and resilience (independent variables) on perceived stress score (dependent variable), entering independent variables in six steps. Normality was graphically assessed using P-P plots, and scatter plots were used to assess homoscedasticity, together indicating linearity of the data. For each set of analyses, the first three models included one independent variable each (PTSD, depression, resilience). The fourth and fifth models in each set included resilience and either PTSD or depression, while the last model in each set included all three independent variables. Adjusted models were estimated that accounted for age, race, ethnicity, and employment status. We conducted two sets of regression analyses: (1) regressions wherein participants were stratified by consent status of sexual initiation and (2) regressions wherein participants were stratified by ever experiencing rape, with and without a covariate accounting for rape perpetrated by an intimate partner. Multicollinearity between predictors was assessed using variance inflection factor (VIF) values, in which a value greater than 5 indicates severe collinearity. We were unable to consider partner perpetration in association with nonconsensual sexual initiation due to insufficient cell sizes.

Theoretical covariates were included in each model, with the exception of education, due to a skewed distribution of responses (many participants were currently enrolled in college), and the sufficiency of income as a marker of socioeconomic status. Stratification was employed rather than interaction terms in an effort to assess comparative relationships between all variables within each stratified group, rather than to assess the joint effect of each independent variable and sexual violence on the outcome. Further, stratification was chosen based on interpretability for practice. Unadjusted coefficients are presented in regression tables. We assessed improvements in model fit at each step by the changes in adjusted R^2^ values, the proportion of the variance in perceived stress score that the independent variables explain collectively. Significance was set at a level of *p* < 0.05.

## 3. Results

### 3.1. Demographic Characteristics

Table 1 presents the demographic profile of participants enrolled in The THRIVE Study. The median age of girls and women (*n* = 65) enrolled in The THRIVE Study was 22 years (IQR: 18–26). Approximately 25% of women identified as Black/African American, 19% were Asian/Pacific Islander, 34% were White, and 34% indicated “Other” race (Often, Hispanic/Latinx; racial identification not mutually exclusive). Forty-two participants (*n* = 27) identified an ethnicity of Hispanic/Latinx. In terms of mental health, 30% (*n* = 19) of women endorsed symptoms consistent with PTSD; mean depression score was 11.3 (SD: 7.6, possible range: 0–28), mean resilience score was 30.0 (SD: 6.2, possible range: 11–40), and mean perceived stress score was 16.2 (SD: 6.9, possible range: 1–32). Regarding sexual violence exposure, of 65 participants, 33 (51%) had ever experienced nonconsensual sexual initiation, 30 (46%) had ever experienced rape.

Compared to those with a consensual sexual initiation, those who experienced nonconsensual sexual initiation were significantly more likely to screen positive for PTSD and had significantly higher depression scores and significantly lower resilience scores (Table 2).

Compared to those who had never experienced rape, those who had experienced rape were significantly more likely to screen positive for PTSD, had significantly higher depression and perceived stress scores, and had significantly lower resilience scores (Table 3).

Depression and perceived stress were significantly positively associated with screening positive for PTSD, while resilience was not associated with PTSD. Depression and resilience were moderately negatively correlated, while depression and perceived stress were weakly-to-moderately positively correlated, and resilience and perceived stress were strongly negatively correlated (Table 4).

### 3.2. Survivors of Nonconsensual Sexual Initiation

In Table 5, regressions are presented by consent status of sexual initiation (nonconsensual (forced, pressured, or unwanted) or consensual (wanted and not pressured or forced)). VIF values (<5) indicated an absence of severe collinearity.

Among women with a history of nonconsensual sexual initiation, symptoms consistent with PTSD were significantly positively associated with perceived stress score across all models (β = 6.514 to 7.523), while depression was significantly positively associated with perceived stress when entered into the model alone (Model 2, β = 0.959, *p* = 0.002) and with resilience (Model 5, β = −0.532, *p* = 0.039). Across all models, resilience was significantly negatively associated with perceived stress score (β = −0.693 to −0.532). Upon the addition of both PTSD and resilience, depression was no longer significantly associated with perceived stress among women who had experienced nonconsensual sexual debut. Among these women, a model including only PTSD symptoms and resilience accounted for the most variance in perceived stress (R^2^ = 0.559), with screening positive for PTSD associated with a 7.4 point higher perceived stress score (β = 7.389, possible range: 0–40, *p* < 0.001), while a 1 point higher reported resilience score (β = −0.687, possible range: 5–50, *p* < 0.001) was associated with a 0.7 point lower perceived stress score.

Among women who experienced consensual sexual initiation, symptoms consistent with depression (β = 0.657 to 0.927) and resilience (β = −0.452 to −0.71) were each significantly associated with perceived stress across all models, while PTSD symptomology was not associated with perceived stress in any models. The most variance in perceived stress was accounted for in the model including depression (β = 0.696, *p* = 0.003) and resilience (β = −0.452, *p* = 0.014) (Model 11, R^2^ = 0.612). In this model, a 1 point higher depression and resilience score were each associated with a 0.7 point higher and a 0.5 point lower perceived stress score, respectively.

### 3.3. Survivors of Rape

In Table 6, multiple linear regressions models are estimated to understand the associations between mental health, resilience, and perceived stress score, stratified by whether women ever experienced rape in their lifetime. VIF values (<5) indicated an absence of severe collinearity.

Among women who had ever experienced rape, depression and resilience were significantly associated with perceived stress in all single-predictor models (Models 1–3), and PTSD, depression, and resilience were significantly associated with perceived stress in all dual-predictor (Models 4–5) models, with PTSD symptoms (β = 5.939, *p* = 0.004) and depression (β = 0.500, *p* = 0.049) positively associated and resilience negatively associated (β = −0.532 to −0.693) with perceived stress. These associations held when partner perpetration of rape was included as a covariate, in all single- (Models 7–9) and dual-predictor (Models 10–11) models. Upon inclusion of PTSD symptoms, depression, and resilience in a single regression, with or without the inclusion of partner perpetration as a covariate (Models 6 & 12), PTSD symptoms (β = 4.582 to 5.075) and resilience (β = −0.624 to −0.626) retained significance, while depression did not. Among survivors of rape, the model accounting for the most variance in perceived stress without accounting for partner perpetration was Model 4 (R^2^ = 0.619), and, when accounting for partner perpetration, Models 10 (R^2^ = 0.633) and 12 (R^2^ = 0.634) accounted for similar amounts of variance in perceived stress. In each of these models (4, 10, and 12), screening positive for PTSD was associated with 4.6 to 5.9 points more perceived stress, while a 1 point higher resilience score was associated with a 0.6 to 0.7 point lower perceived stress score. Experience of partner-perpetrated rape was not significantly associated with perceived stress score in any regression.

Among women who never experienced rape, PTSD symptoms were not associated with perceived stress in any regression model, while depression (β = 0.515 to 0.694) and resilience (β = −0.421 to −0.563) were positively and negatively associated with resilience in single-, dual-, and three-predictor models, respectively. Model 18, including three predictors, accounted for the most variance in perceived stress among women who had never experienced rape (R^2^ = 0.604). In Model 18, a 1 point higher depression score was associated with a 0.52 point higher perceived stress score, while a 1 point higher resilience score was associated with a 0.42 point lower perceived stress score.

## 4. Discussion

This study examined the associations of symptoms consistent with PTSD, depression, and resilience with perceived stress, stratified by experiences of nonconsensual sexual initiation and lifetime experience of rape among women. Results of this analysis indicate four key findings. First, symptoms consistent with PTSD played a significant role in perceived stress among all women who experienced sexual violence, irrespective of type of sexual violence experience, while depression was associated with perceived stress among their consensual sexual initiation counterparts and women who had not experienced rape. Second, resilience was consistently associated with less perceived stress among all women, accounting for symptoms consistent with PTSD and depression. Third, the attenuating association of resilience with perceived stress appeared stronger among women who experienced sexual violence than women who did not. Finally, counter to previous findings in the literature, perpetration of rape by an intimate partner was not significantly associated with perceived stress when accounting for PTSD and depression. We discuss these findings in the context of extant literature on stress, mental health, and resilience among women survivors of sexual violence.

Symptoms consistent with PTSD play a significant role in the perceived stress score of all survivors of sexual violence in our sample, wherein PTSD symptoms were significantly associated with greater perceived stress in adjusted models, irrespective of type of sexual violence. Among survivors, PTSD retained a significant association with perceived stress when accounting for depression, while depression did not remain significant. This finding is particularly significant, given that nearly one-third (30%) of our sample of women reported symptoms consistent with PTSD. By contrast, depression was significantly associated with perceived stress among women who did not experience the respective types of sexual violence, including when accounting for PTSD, indicating the importance of services tailored to survivors of sexual violence in relation to PTSD in particular. A range of traumatic events and childhood adversities, aside from sexual violence, can play an important role in PTSD among women, and our study revealed that symptoms consistent with PTSD significantly enhance perceptions of stress among women who experienced sexual violence. U.S. women are two to three times as likely to develop PTSD as men [63,64]; while sexual violence may contribute to this, it may also compound upon existing PTSD among women, further increasing stress and the negative sequalae of sexual violence. Moreover, mental health services are routinely underutilized, and survivors of sexual violence are likely to avoid care-seeking due to fear of re-traumatization [65]. Mental health conditions are likely to worsen if left untreated, often leading to additional adverse health outcomes and compensatory behaviors, including substance use and suicide [66,67].

Among our sample, resilience was significantly associated with lower perceived stress among all women, regardless of sexual violence exposure. However, the attenuating association of resilience with perceived stress appeared stronger in magnitude among women who experienced rape or nonconsensual sexual initiation than women who did not. These findings shine light on the importance of nurturing resilience among all women, especially survivors of sexual violence, which has also been shown to be protective against the development of PTSD in the aftermath of trauma [68]. We identify that resilience may be a critical tool for reducing perceived stress among women with experiences of sexual violence, who may be more vulnerable to the negative effects of perceived stress on mental health conditions compared to unabused women [69]. While screening positive for PTSD is significantly associated with a large increase in perceived stress, the breadth of possible scores in the measure of resilience, coupled with the statistically significant attenuation of perceived stress by resilience, indicate that augmented resilience can outpace the negative impacts of screening positive for PTSD. Despite the importance of resilience as a positive coping strategy, there are relatively few interventions that aim to promote resilience among women who have experienced sexual violence [70]; instead, interventions often address PTSD and trauma-related stress with strategies such as Cognitive Behavioral Therapy. Although most cognitive behavioral approaches focus their treatment on factors related to a traumatic memory, effective PTSD treatments have shown that trauma-focused treatment to promote resilience may be more effective than trauma-focused therapy [71]. Additionally, carefully tailored interventions that promote resilience can play a vital role in recovery following trauma [72]. Appropriate integration of resilience into interventions may be impactful against the robust negative effects of symptoms consistent with PTSD, depression, and sexual violence on stress in women.

Finally, the results of this analysis indicate no association between experiencing perpetration of rape by an intimate partner and perceived stress when accounting for PTSD, depression, and resilience. While previous research has identified associations between intimate partner perpetration of violence and PTSD, depression, anxiety, and other outcomes [44,45,46], our analyses cannot corroborate these findings nor suggest meaningful influence of partner-perpetrated sexual violence on perceived stress.

### 4.1. Strengths and Limitations

The present study was limited by several features that are common in sociobehavioral research, including self-reported measures of mental health, which can result in social desirability bias; cross-sectional data, which limit the ability to draw causal inferences; and recall bias, as it relates to recollection of sexual violence and first intercourse. These factors should be considered when evaluating the extent to which the findings can be generalized. While the sample was racially and ethnically diverse, its small size precluded the ability to examine racial and ethnic differences that likely exist. Future research should examine such differences, given high national prevalence estimates of sexual assault and PTSD among Black, Latinx, and Indigenous populations [73]. The age range of the sample was limited, precluding the ability to examine potential observed associations for children younger than 18 and women older than 45. Further, age of participants skewed younger, limiting the ability to generalize results to women 30–45. We were unable to determine whether symptoms consistent with PTSD and depression were directly attributed to sexual violence history or some other type of trauma over the life course. Likewise, we acknowledge that resilience is a dynamic process that we measured as a snapshot in time [74]. Moreover, we did not measure underlying variables that may have contributed to resilience, such as social connections with others and help-seeking behaviors. Future research should integrate qualitative research methods to elucidate these unmeasured factors to facilitate the development and implementation of interventions that promote resilience to mitigate the negative effects of adverse mental health. Finally, we acknowledge that the data are insufficient to assess the severity of experiences of sexual violence but rather assess whether participants conceptualize themselves as having experienced each type of sexual violence queried. This use of reported experiences kept the data and voices of women in the study as reflective of actual experiences as possible. Similarly, experiences of sexual violence other than those reflective of subgroup eligibility were not controlled for in analyses. Understanding the total mental health burden of women enrolled in the study associated with lifetime experiences of sexual violence allows for better responsiveness and anticipation of mental health outcomes based on the information likely to be provided to practitioners and public health interventionists. Similarly, there are significant associations between several of the independent variables, indicating the potential for multicollinearity [75]. However, VIF values indicated that no collinearity was severe (VIF > 5) [75].

Additional notable strengths are also worth discussion. On average, women enrolled in the study were younger, which decreased the likelihood of recall bias, as it pertains to assessment of lifetime history of rape and sexual assault. The sample was ethnically diverse, and over two-thirds of the sample were recruited through study advertisements on social media platforms, both of which increase the generalizability of study findings, as women who did not seek services or who are traditionally underrepresented in research were identified and enrolled.

### 4.2. Future Directions

Our study findings have several implications for future research and clinicians working with survivors of sexual violence. Through cross-sectional research, we found negative associations between resilience and perceived stress and positive associations between symptoms consistent with PTSD and perceived stress among survivors of nonconsensual sexual initiation and rape. Future analyses should investigate sustained and causal relationships between symptoms consistent with PTSD, resilience, and perceived stress among survivors of sexual violence and whether or not this relationship is consistent across sexual and gender identity. Additionally, qualitative research studies that elucidate the context of individual and collective resilience and coping strategies in the midst of experiencing PTSD symptoms is needed to inform interventions for survivors of sexual violence. Current trauma-focused interventions have been shown to both reduce PTSD symptoms and improve resilience [76]; however, to our knowledge, there is a lack of such interventions designed and tested among survivors of sexual violence. The high proportion of women experiencing sexual violence stresses the importance of promoting resilience while mitigating symptoms consistent with PTSD beyond in-person support to innovative methods such as therapist-assisted online support [77].

## 5. Conclusions

Overall, this study provides support for leveraging resilience and addressing symptoms consistent with PTSD to reduce perceived stress among women with lifetime experiences of sexual violence, as well as among women without such experiences. Among survivors of sexual violence, irrespective of type, PTSD and resilience were robustly associated with perceived stress. Resilience should be of considerable interest among violence researchers in search of identifying efficacious responses to traumatic stress and targets of PTSD prevention and treatment among sexual violence survivors.

## Figures and Tables

**Table 1 ijerph-19-04796-t001:** Study Participant Characteristics, Mental Health, and Sexual Violence Exposure, San Diego, CA 2018 to 2020 (*n* = 65).

Variables	*n* (%)
Age in years, median (IQR)	22 (18, 26)
Race (Not Mutually Exclusive)	
Black/African American	16 (24.6)
White	22 (33.8)
Asian/Pacific Islander	12 (18.5)
Other	22 (33.8)
Ethnicity	
Hispanic/Latinx	27 (41.5)
Educational Attainment	
High School Diploma, GED, or Less	36 (55.4)
Some Trade, Vocational School, or College or more	28 (43.1)
Employment Status	
Unemployed	20 (30.8)
Employed Part-Time	29 (44.63)
Employed Full-Time	15 (23.1)
Mental Health	
Symptoms consistent with PTSD	19 (29.2)
Depression, mean (SD), possible range: 0–28	11.3 (7.6)
Resilience, mean (SD), possible range: 11–40	30.0 (6.2)
Perceived Stress, mean (SD), possible range: 1–32	16.2 (6.9)
Sexual Violence Exposure	
Nonconsensual Sexual Initiation	33 (50.8)
Partner	31 (47.7)
Non-Partner	2 (3.1)
Ever Raped	30 (46.9)
Partner	17 (26.2)
Non-Partner	13 (20.0)

IQR, interquartile range; GED, general equivalency diploma; PTSD, post-traumatic stress disorder; SD, standard deviation.

**Table 2 ijerph-19-04796-t002:** Bivariate Associations between Mental Health Outcomes, Resilience, and Perceived Stress by Sexual Initiation Consent Status, San Diego, CA 2018 to 2020 (*n* = 65).

	Nonconsensual Sexual Initiation (*n* = 33)	Consensual Sexual Initiation (*n* = 32)	*p*
PTSD, *n* (%)	14 (42.42%)	5 (15.6%)	0.017
Depression, Mean (SD)	13.82 (7.80)	8.65 (6.45)	0.002
Resilience, Mean (SD)	28.85 (6.81)	31.19 (5.36)	0.003
Perceived Stress, Mean (SD)	18.61 (6.68)	13.71 (6.24)	0.066

**Table 3 ijerph-19-04796-t003:** Bivariate Associations between Mental Health Outcomes, Resilience, and Perceived Stress by Ever Experience of Rape, San Diego, CA 2018 to 2020 (*n* = 65).

	Ever Rape (*n* = 30)	Never Rape (*n* = 35)	*p*
PTSD, *n* (%)	16 (53.33%)	3 (8.57%)	<0.001
Depression, Mean (SD)	14.87 (7.04)	8.18 (6.67)	<0.001
Resilience, Mean (SD)	27.90 (6.71)	31.82 (5.17)	<0.001
Perceived Stress, Mean (SD)	19.47 (6.27)	13.38 (6.14)	0.005

**Table 4 ijerph-19-04796-t004:** Bivariate Associations between Mental Health Outcomes, Resilience, and Perceived Stress, San Diego, CA 2018 to 2020 (*n* = 65).

	Positive PTSD Screen	Negative PTSD Screen	*p*
Depression, Mean (SD)	18.21 (6.15)	8.40 (6.09)	<0.001
Resilience, Mean (SD)	28.58 (6.48)	30.58 (6.07)	0.121
Perceived Stress, Mean (SD)	21.16 (6.01)	14.16 (6.16)	<0.001
	**Pearson Correlation Coefficient, *r***		** *p* **
Depression × Resilience	−0.391		0.001
Depression × Perceived Stress	0.548		<0.001
Resilience × Perceived Stress	−0.687		<0.001

**Table 5 ijerph-19-04796-t005:** Multiple Linear Regression Models of Independent Variables (PTSD, Depression, Resilience) on Perceived Stress Stratified by Consent Status of Sexual Initiation (*n* = 65).

	Nonconsensual Sexual Initiation (*n* = 33)						
	PTSD, β	*p*	Depression, β	*p*	Resilience, β	*p*	R^2^
Model 1	**7.523**	**0.017**					−0.10
Model 2			**0.959**	**0.002**			0.161
Model 3					**−0.693**	**<0.001**	0.334
Model 4	**7.389**	**<0.001**			**−0.687**	**<0.001**	0.599
Model 5			**0.551**	**0.039**	**−0.532**	**0.002**	0.429
Model 6	**6.514**	**0.003**	0.306	0.175	**−0.599**	**<0.001**	0.616
**Consensual Sexual Initiation (*n* = 32)**							
Model 7	**6.658**	**0.038**					0.183
Model 8			**0.927**	**<0.001**			0.504
Model 9					**−0.71**	**0.001**	0.427
Model 10	4.225	0.199			**−0.652**	**0.003**	0.446
Model 11			**0.696**	**0.003**	**−0.463**	**0.014**	0.612
Model 12	1.800	0.530	**0.657**	**0.007**	**−0.452**	**0.019**	0.601

All regressions are adjusted for age, race, ethnicity, employment status, and case/control status. **Bolded** effects are significance at a level of *p* < 0.05.

**Table 6 ijerph-19-04796-t006:** Multiple Linear Regression Models of Independent Variables (PTSD, Depression, Resilience, Partner Perpetration) on Perceived Stress Stratified by Ever Rape (*n* = 65).

	Ever Rape								
	PTSD, β	*p*	Depression, β	*p*	Resilience, β	*p*	Partner Perpetration, β	*p*	R^2^
Model 1	4.817	0.125							−0.054
Model 2			**0.904**	**<0.001**					0.304
Model 3					**−0.659**	**<0.001**			0.434
Model 4	**5.939**	**0.004**			**−0.698**	**<0.001**			0.619
Model 5			**0.500**	**0.049**	**−0.483**	**0.006**			0.517
Model 6	**5.075**	**0.030**	0.193	0.450	**−0.624**	**<0.001**			0.611
Model 7	4.718	0.144					−0.777	0.770	−0.105
Model 8			**0.942**	**<0.001**			−2.356	0.270	0.314
Model 9					**−0.685**	**<0.001**	−2.476	0.178	0.456
Model 10	**5.715**	**0.005**			**−0.718**	**<0.001**	−2.016	0.202	0.633
Model 11			**0.530**	**0.032**	**−0.502**	**0.003**	−2.783	0.113	0.558
Model 12	**4.582**	**0.044**	0.247	0.327	**−0.626**	**<0.001**	−2.250	0.162	0.634
**Never Rape**									
Model 13	7.063	0.054							0.418
Model 14			**0.694**	**0.009**					0.489
Model 15					**−0.563**	**0.008**			0.491
Model 16	5.679	0.087			**−0.507**	**0.013**			0.532
Model 17			**0.563**	**0.021**	**−0.459**	**0.019**			0.578
Model 18	4.770	0.119	**0.515**	**0.030**	**−0.421**	**0.027**			0.604

All regressions are adjusted for age, race, ethnicity, employment status, and case/control status. **Bolded** effects are significance at a level of *p* < 0.05.

## Data Availability

Data are not available due to the sensitive nature of the study.

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
