# Peer review of "The Effect of Adverse Mental Health and Resilience on Perceived Stress by Sexual Violence History"

_ijerph, 2022, doi:10.3390/ijerph19084796_

Round 1

Reviewer 1 Report

Dear Authors; this is a successful study on the association of adverse mental health an resilience with perceived stress. Its presentation at the current status is 2 degrees under journal standards and needs "some serious" extra work. Regards,

P.S.

[1] Writing:

1-1 Abstract: In line 18-22 you gave too much descriptive statistics info and buried the key results beta values and p-values in lines 22-24. Please make the lines 18-22 more concise and shift the attention to the lines 22-24. Expand lines 22-24.

1-2 One key limitation on this study and a "gold opportunity" for the second follow-up paper is its limitation on gender. It has focus on female victims and no investigation of male victims. Please add this issue in the future work paragraph in the discussion section.

1-3 This journal is international and is being read across planet. The authors are recommended to add a list of used abbreviations in the manuscript right before the reference section for the non-North American readers easy access. Example:

Abbreviations

IPV:  Intimate Partner Violence

[2] Statistical:

2-1 Please explain how did you check normality of your outcome variables in your regression model. Add details to the manuscript.

2-2 Please explain how did you check linearity of relationship between covariates and the outcomes in your modelling. Did you use boxcox test ? Add details to the manuscript.

2-3 Lines 189-192 miss the formula of the regression models. Please add them.

2-4 The current presentation is impotent to deliver its key message to the reader in one shot. Please, add a Figure.1 right after Table.3. In this figure, add two curves presenting the relationship between PSS versus resilience in models 6,12 (the covariates PTSD and Depression Score are averaged). Give comments to the Figure.1.  Legend: Nonconsensual/consensual 

2-5 Same as item 2-4 this time for a second Figure.2. after Table.4. with models 6,18. Legend: Everrape/Neverrape

2-6 The authors may add other important Figure scenarios that those mentioned in item 2-4 and item 2-5.   

Reviewer 2 Report

Dear Editor and authors,

This research examines resilience as an intervention tool related to mental health in women with sexual violence history and partner perpetration. I considered that deals with global issues because the research improves the information about mental health and intimate partner violence.

 Some issues that they should consider addressing before publication are described:

In the abstract, the PTSD abbreviation should be described.

In the methods section, the authors indicated the eligible participants, but the inclusion criteria of the study should be indicated in this section. Although the alpha Cronbach was indicated for each instrument; it is not specified whether the values have been calculated for the study sample. Additionally, I recommend that the authors consider calculating the alpha ordinal for their sample.

In the results section, I suggest that the results of Table 2 would be shown separately. A table with the comparisons between nonconsensual and consensual sexual initiation, and the comparisons between ever rape and never rape. And, another table with the association between depression and perceived stress with positive or negative PTSD Screen. In addition, I recommend that beta would be shown with the p-value. For example, “PTSD were significantly positively associated with perceived stress score across all models (β = 6.514–7.523; p = XXXX)”. Also, the authors should indicate what they mean by range. My suggestion is that no significant results should be included in Table 3 and Table 4. In the text, the beta should be indicated when the association was mentioned.

In the discussion and limitation sections, the age range was indicated like 14-45; however, the description of the sample indicated that women aged18 and older expressed interest in the study.

Reviewer 3 Report

Review

The Effect of Adverse Mental Health and Resilience on Perceived Stress by Sexual Violence History

Abstract

The present research aimed to explore the possible role of resilience and adverse mental health on perceived stress among survivors of sexual violence. The study involved 64 women of age 18 or older (median age was 22 years), among which 33 had experienced non-consensual sexual initiation while 30 had experienced rape. From data analysis PTSD resulted to be significantly associated with perceived stress among participants, while, on the contrary, resilience was associated with lower perceived stress.  Therefore, the research results highlight the importance of taking resilience and PTSD into consideration for future interventions with women who have been victims of sexual violence.   

Authors’ Abstract

The article abstract is well organized and allows the reader to adequately understand the manuscript content. However, the specific aim of the research could be expressed more clearly and in detail to allow the readers to immediately understand it. Moreover, in an abstract it is not necessary to indicate the descriptive statistics such as standard deviation, mean and range for depression, resilience, and perceived stress, while the authors should instead focus on better specifying the instruments used to assess such variables.  

Introduction

In the article Introduction section, the authors explore the study background and present an overview of the findings already available concerning the research themes. The section appears to be very well organized, with different sub-sections that significantly contribute to offer the readers a detailed perspective on the research themes. Therefore, the study background is well described, clear and interesting.

Moreover, another commendable aspect is that every theme that emerges in the section is presented together with adequate references to the current literature available, which is particularly important for scientific research in which the source of every information should always be traceable.      

A possible suggestion that could be given to make the section even more complete is related to the sub-section “1.1. PTSD, Depression, and Stress among Survivors of Sexual Violence”, on page n. 2, and more specifically to the sentence:

“Survivors of sexual violence have a heightened level of perceived stress [29], which is associated with predictability or control of one’s life, extent of life changes, and one’s ability to deal with problems or difficulties”.

The authors should consider the possibility to describe a little more the relation between the different aspects mentioned here (predictability or control of one’s life, extent of life changes, and one’s ability to deal with problems or difficulties) and a heightened level of perceived stress among survivors of sexual violence, perhaps briefly reporting some specific results in the scientific literature available on the theme.

Materials and Methods    

In the Materials and Methods section, the authors describe in detail the methodology and procedures they have adopted for data collection and analysis. 

The section, similarly to the previous one, appears to be generally well organized. Both the research design and measures are described in detail (with even some examples of the measures items, which is commendable), and the adequate references to the related scientific literature are inserted.

However, the meaning of the title of the first sub-section, “Parent study” (page n. 3) is not completely clear, therefore the authors should consider the possibility to change it with one that could better summarize the sub-section content.

Another aspect the authors should consider is the fact that the precise timeline of the present study is not completely clear, especially considering the distinction between the baseline study and the subsequent one, therefore the authors should provide a more detailed description of every step they implemented for the present research (for example, how participants were contacted, when questionnaires were administered, if it was in a different time from the original study etc.).

Furthermore, concerning instead more specifically the Measures sub-section, the authors are advised to remove the classification in their description of the different measures (dependent variable, independent variables, stratification variables), which should be more appropriately explained in the paragraph dedicated to statistical analysis. As regards instead the present sub-section, the authors could more specifically create only two paragraphs: one for the variables related to mental health indices (Perceived Stress, PTSD, Resilience, Depression) and one for the variables related to Sexual Violence History (sexual initiation, experiencing rape and experienced rape perpetrated by an intimate partner).

Lastly, concerning the statistical analysis employed, it should be noted that the term “Stepwise multiple linear regressions” (which is presented for the first time in the manuscript in the Statistical Analysis sub-section, on page n. 4 line 189)  is not completely adequate, since there is no selection of predictors with the precise stepwise technique in the current study. Separate regression models are conducted by including the predictors first one at a time (mod1-mod3), then two at a time (mod4-mod5) and then all three together (mod6), therefore the authors should use a different term to describe the implemented process. 

Results  

In the Results section, the authors present their research fundamental results obtained from the data analysis. The section, like the previous ones, appears to be generally well organized.

However, there are some modifications that should be applied. More specifically, in Table 2 (page n. 6) many different data are present, which could result overwhelming for readers, therefore the authors should consider the possibility to organize the results in two different tables, one for the relations between mental health indices and Sexual Violence History (first two parts of Table 2) and another for the relations between the different mental health indices (last two parts of Table 2). Moreover, the association between mental health indices and intimacy partner perpetrated rape appears to be missing, therefor the authors should add it as well.  

Lastly, as regards the correlation between the mental health indices, the authors should refer to Pearson's correlation coefficient r for PTSD as well, an aspect that appears to be missing.

Another aspect that needs revision is the fact that in Table 2 and Table 3 (page n. 7) there are numbers that do not add up for “Consensual sexual initiation” (the reported number is 32, however according to the total number of participants it should be 31) and “Never Rape” (number reported = 35, it should probably be 34) to arrive at the sample total (N = 64). The authors should therefore revise the numbers provided in the two Tables and correct eventual mistakes.  

In the Results section there are other elements to reconsider as well, for example in the line 259 (page n. 7) an incorrect reference is made to Model 5, while it should probably be Model 11 instead. Moreover, in line 260 (page n. 7) the authors refer to an increase of 0.6 points in stress for a unitary increase in depression but being beta = 0.696 it would be more appropriate to indicate it as equivalent to 0.7.

Furthermore, in the note of Table 3 (on page n. 7), the authors state that “All regressions are adjusted for age, race, ethnicity, employment status, and case / control status”, however the "educational attainment" variable, which is present in Table 1, was not mentioned and thus it appears it was not included in the regression analysis as a control variable. The authors should therefore explain why this variable was not included.

Moreover, in Table 4 (on pages n. 7-8) the R-squared values ​​of Model 1 and Model 7 are missing; therefore the authors should insert them. 

Lastly, in line 292 (on page n. 8) the authors refer to Model 19, however, this is not present in Table 4, therefore the authors probably meant Model 4.

Discussion

In the Discussion section, the authors present an overview of their research findings, their possible meaning, and practical implications, and compare them with recent literature available.

The section appears to be well written and detailed, while the discussed results are very interesting. The references to the scientific literature available are adequate and allow the readers to consider the research findings in a broader perspective.

Moreover, the possible implication for future research and interventions to support women who are victims of sexual violence are well presented and significant.

However, an aspect that should be reconsidered by the authors is the use of the expression “women of color” (in the line 381, page n. 10) since it could be considered rather inadequate and vague, and therefore the authors should consider the possibility to replace it with a more appropriate one.  

Lastly, the study limits are presented in detail as well, a commendable aspect that is not always present in similar research and that confirms the authors’ precise attention to every aspect of the study. However, it might be more effective for a smoother reading to write the limits and the suggestions for the future research in a separate section.

Conclusions       

In the Conclusions section the authors briefly review their study and offer final insights into their research. The section appears adequate and well-written, however, the authors are advised to be particularly mindful of the words used to describe the obtained research results, since terms such as “demonstrate” (on line 437, page n. 11) could be considered not completely adequate, considering that the statistical analysis employed only allow to obtain a certainty concerning the correlation between variables, which however does not equal to a causal relation, as the term “demonstrate” could suggest.

However, the possible misconception the term “demonstrate” could arise does not affect the rest of the manuscript and in particular the Results and Discussion sections, in which the authors adequately describe the correlation between variables, without attributing to it a causal meaning. 

The authors should therefore only change the term used in the present section, to further avoid misunderstandings.

References

The study references provided, both in the text and in the References section, appear to be adequate.

Use of English

The English language of the text is perfectly adequate and correct, the sentences are written in a way that is easy to understand for every reader.   

Overview

Overall, the article appears to be well-written and organized, with adequate references to the current scientific literature, and it explores some very interesting themes. Only a few minor elements in the different sections of the manuscript should be reviewed in order to make it even more suitable for publication, in particular in the Materials and Methods and in the Results section.  

Round 2

Reviewer 1 Report

Dear Authors, most of my concerns were addressed satisfactorily. Regards.